# Can 17 hydroxyprogesterone caproate (17P) decrease preterm deliveries in patients with a history of PMC or pPROM?

Gal Cohen[1,2]*, Maya Shavit[1,2], Netanella Miller[1,2], Rimon Moran[1,2], Yael Yagur[1,2], Omer Weitzner[1,2], Michal Ovadia[1,2], Hanoch Schreiber[1,2], Gil Shechter-Maor[1,2], Tal Biron-Shental[1,2]

1 Department of Obstetrics and Gynecology, Meir Medical Center, Kfar-Saba, Israel, 2 Sackler Faculty of Medicine, Tel Aviv University, Tel Aviv, Israel

* galcwork@gmail.com

## Abstract

### Background

A history of spontaneous preterm birth (sPTB) is a significant risk factor for recurrence. Intra-muscular-7α-hydroxyprogesterone caproate (17P) has been the preventive treatment of choice until the recent "Prolong study" that reported no benefit.

### Objective

To determine the benefit of (17P) treatment in preventing reoccurrence of sPTB, by evaluating two presenting symptoms of the first sPTB: premature contractions (PMC) and preterm premature rupture of membranes (pPROM).

### Study design

This retrospective study included 342 women with a previous singleton sPTB followed by a subsequent pregnancy. sPTB were either due to PMC (n = 145) or pPROM (n = 197). During the subsequent pregnancy, 90 (26.3%) patients received 250 mg 17P IM. Each presenting symptom–PMC or pPROM–was evaluated within itself comparing treated vs. untreated groups. Data were analyzed using t-test, Chi-square and Fisher's exact test. Logistic regression analysis was also performed.

### Results

Patients treated with 17P in the subsequent pregnancy had delivered earlier in the previous pregnancy (33.4w vs. 35.3w in the PMC group, and 34.1w vs. 35.7w in the pPROM group, p<0.001). In the following pregnancy, they had higher admission rates due to suspected preterm labor (31.7% vs. 10.9% in the treated vs. untreated PMC group (p = 0.003) and 26.1% vs. 5.4% in the treated vs. untreated pPROM group (p<0.001). In both groups, but more prominently in the previous PMC group, treatment compared to non-treatment in the subsequent pregnancy significantly prolonged it (4.3w vs. 2.6w in the PMC group (p = 0.007), and 3.7w vs. 2.7w in the pPROM group (p = 0.018)).

**Data Availability Statement:** The data underlying the results presented in the study are available and are attached under "supporting information" File name: S1_raw_data.

**Funding:** The authors received no specific funding for this work.

**Competing interests:** The authors have declared that no competing interests exist.

**Abbreviations:** PMC, Premature contractions; pPROM, Preterm premature rupture of membranes; sPTB, Spontaneous preterm birth; PTB, Preterm birth; GA, Gestational age; 17P, 17 Hydroxyprogesterone caproate; CD, Cesarean delivery; GDM, Gestational diabetes mellitus; PET, Pre-eclampsia; HTN, Hypertension; IUGR, Intrauterine growth restriction; SGA, Small for gestational age; MFM, Maternal fetal medicine.

The presenting symptom of sPTB in the following pregnancy tended to recur in cases of another sPTB, with a significantly greater likelihood of repeating the sPTB mechanism in cases with PMC, regardless of receiving 17P (69% in the PMC cohort and 60% in the pPROM cohort, p<0.001).

## Conclusions

17P might delay preterm delivery in patients with a previous sPTB on an individual level (prolongation of the pregnancy for each patient compared to her previous delivery). Therefore, our results imply that 17P can decrease potential premature delivery complications for patients with a previous sPTB due to PMC or pPROM.

## Introduction

Preterm birth (PTB) accounts for 12.5% of births annually, and is a major cause of neonatal morbidity and mortality [1]. Mechanisms leading to preterm birth include: Induction of labor due to maternal or fetal indications, accounting for 30–35% of PTB; premature contractions (PMC) with intact membranes, accounting for 40–45% of PTB; and preterm premature rupture of the membranes (pPROM) accounting for 20–25% of PTB [2]. Births that follow PMC and pPROM are also called spontaneous preterm births (sPTB).

Women with a history of sPTB have a 2.5-fold increased risk for another sPTB in the subsequent pregnancy [3]. This risk is increased by 2.2-fold when the presenting symptom was PMC, and by 3.3-4-fold when the presenting symptom was pPROM [3,4].

Chorioamnionitis or placental abruption might also be involved in the mechanism of PTB. In these cases, maternal fever or vaginal bleeding, respectively, accompany the presenting symptom of PMC or pPROM. However–these ethnologies are relatively rare and are associated with lower rates of recurrence [5,6].

Given the significant adverse outcomes associated with PTB, attempts to reduce the risk of recurrence have been made. Progesterone has been found to inhibit myometrial contraction [7], and so, one strategy was the use of supplemental progestogens, including intramuscular (IM) 17α-hydroxyprogesterone caproate (17P) in women with a singleton pregnancy and a history of singleton spontaneous PTB. This treatment was evaluated in two large randomized-controlled-trials. Meis et al. [8] demonstrated a substantial decrease in the rate of recurrent sPTB with 17P treatment compared to placebo, among high-risk populations, as well as a decrease in neonatal adverse outcomes such as necrotizing enterocolitis, intraventricular hemorrhage, and need for supplemental oxygen. This made 17P the preventive treatment of choice by the American College of Obstetricians and Gynecologists (ACOG) and the Society for Maternal-Fetal Medicine (SMFM), until the recent PROLONG study was published, showing 17P had no benefit in decreasing rates of recurrent PTB or neonatal morbidity [9].

On July 2020, the SMFM published a statement regarding the two trials,[7] suggesting the difference in the results was attributable to the difference in populations, as the population included in the Meis study[5] had a higher baseline risk for recurrent PTB compared to that in the PROLONG study.[6] They concluded that it is reasonable for providers to use 17P in women with a profile similar to the very high-risk population reported in the Meis trial.

Vaginal progesterone also has been studied for the indication of prior PTB. A systematic review and meta-analysis [10] comparing the relative effects of different kinds of progesterone in preventing PTB, concluded that both vaginal progesterone and 17P were effective at reducing the risk for recurrent PTB.

On August 2021, the ACOG stated that patients with a singleton pregnancy and a prior sPTB should be offered progesterone supplementation (either vaginal or IM) in the context of a shared decision-making process incorporating the available evidence and patient's preferences [11].

The argument regarding 17P has not been settled yet, since there are no other promising alternatives to prevent sPTB and safety data allow 17P [12]. Therefore, there is a real need for more studies that will determine the specific populations or situations in which 17P might have a treatment advantage.

Neither of the two trials mentioned above evaluated the efficacy of 17P according to the presenting symptom of the prior sPTB. This was pointed out by Gonzalez-Quintero [13], who showed higher rates of recurrent sPTB in the population with prior PMC compared to pPROM. Another retrospective study found similar results [14]. These trials only included women who were treated with 17P in the subsequent pregnancy, leaving the question of its efficacy compared to placebo for each of the two presenting symptoms, unresolved.

Moreover, studies evaluating 17P have always estimated its efficacy by demonstrating decreased rates of recurrent PTB at different thresholds (<37w, <35w, <34w, or <32w). None evaluated the individual benefit of 17P in prolonging the subsequent delivery, for each woman relative to herself.

The current study evaluated the individual benefit of 17P treatment in preventing recurrent sPTB in the face of PMC and pPROM as the two main presenting symptoms.

## Methods and materials

This retrospective cohort study included women with two consecutive singleton deliveries at a single academic institution from January 2014 to January 2020, when the first delivery was a sPTB <37w with a presenting symptom of either PMC or pPROM.

We divided the cohort into 2 groups according to the presenting symptom of the preceding PTB: PMC or pPROM. Each group was further divided into 2 subgroups: treated with 17P in the subsequent pregnancy–including all women treated with weekly 250 mg 17P IM starting at 16 weeks of gestation, and untreated—those not treated with any kind of progesterone (vaginal, PO, IM) or any other medication to prevent sPTB in the subsequent pregnancy.

Data were retrieved using the electronic maternal database of the delivery room. Information regarding pregnancy surveillance was retrieved, exploring each woman's electronic medical records during hospitalizations and as an outpatient. These included maternal demographics and characteristics (age, ethnicity, gravidity, parity, background diseases, BMI, smoking), previous cesarean deliveries (CD), and previous cervical interventions including dilation and curettage, dilation and evacuation, conization, and surgical hysteroscopy.

Preceding pregnancy and delivery characteristics were also collected, including diabetes (pre-gestational or gestational), intrauterine growth restriction (IUGR), hypertension (HTN) or pre-eclampsia (PET), progesterone treatment of any kind (200 mg a day vaginally, 250 mg IM weekly, 400 mg a day PO), gestational age (GA) at delivery and mode of delivery.

Subsequent pregnancy and delivery characteristics were also retrieved, including interpregnancy intervals, diabetes (pre-gestational or gestational), IUGR, HTN or PET, high-risk surveillance, admissions to hospital in fear of PTB, cervical lengths during pregnancy (14-16w, 20-24w and at admission for delivery), progesterone treatment of any kind as mentioned, GA at delivery, rates of PTB <32, 35, 37w and mode of delivery.

The data collected regarding neonatal outcomes in the subsequent delivery included birth weight, rate of small for gestational age (SGA) neonates, rate of low birth weight (LBW) defined as <2,500g, rate of very low birth weight (VLBW) defined as <1,500g and 5-minute Apgar scores.

Gestational or pregestational diabetes were defined as glucose challenge test score $\geq 200$, one or more pathological values in oral glucose tolerance test scores with thresholds defined according to ACOG guidelines, or a known diagnosis of diabetes mellitus noted in the patient's medical records.

High-risk surveillance was defined as pregnancy surveillance by a designated Maternal-Fetal Medicine (MFM) specialist in a high-risk clinic.

Birth weight percentiles were calculated using local, gender-specific, population-based, birth weight curves [15].

Given the inherent differences between the treated and untreated populations in our cohort, with the treated population initially having a higher risk profile, we tried to overcome the selection bias of the treated population by evaluating the **individual** benefit of 17P treatment for each woman, demonstrated by the delta between GA at the subsequent delivery and at the first preterm birth.

Primary outcomes were the rates of recurrent sPTB at $<37$, $<35$, $<32$w gestation in each group and difference in GA between subsequent delivery and prior sPTB for each woman. Secondary outcomes were characteristics of the subsequent pregnancy: cervical length, rate of maternal fetal medicine (MFM) specialist surveillance and hospital admissions for premature uterine contractions and suspected sPTB.

### Ethics approval

The study was approved by the Institutional Ethics Committee–The Meir Medical Center Ethics Committee, in August 2019, approval number 0369-19-MMC. Since the study was based on patient records, the Meir Medical Center Ethics Committee waive the need for informed consent. All methods were performed in accordance with the relevant guidelines and regulations.

### Data analysis

Continuous variables were evaluated by their distribution using the Shapiro-Wilk test. Variables with a normal distribution were presented by mean ± SD, and were compared using t-test. Categorical data were compared using Chi-squared or Fisher's exact test, each when appropriate. Multivariate logistic regression analysis and adjusted odds ratios (OR) were calculated to find variables that had an independent influence on preterm birth $<35$w. A probability value of $p < 0.05$ was considered significant. All analyses were performed using SPSS-25 software (IBM, Armonk, NY, USA).

## Results

During the study period, 592 women had consecutive deliveries in our institution, with the first delivery a sPTB $<37$w. Of these, 342 women met the inclusion criteria; 145 had a PTB due to PMC (group 1) and 197 due to pPROM (group 2). Fig 1 shows the details of the study population. S1 Table shows the inclusion and exclusion criteria for the study.

### Comparison between patients with previous sPTB due to PMC, treated or not treated with 17P during the subsequent pregnancy

The groups had similar basic characteristics, including maternal age, ethnicity, background diseases, BMI, smoking, nulliparity, previous CD and previous cervical interventions (Table 1).

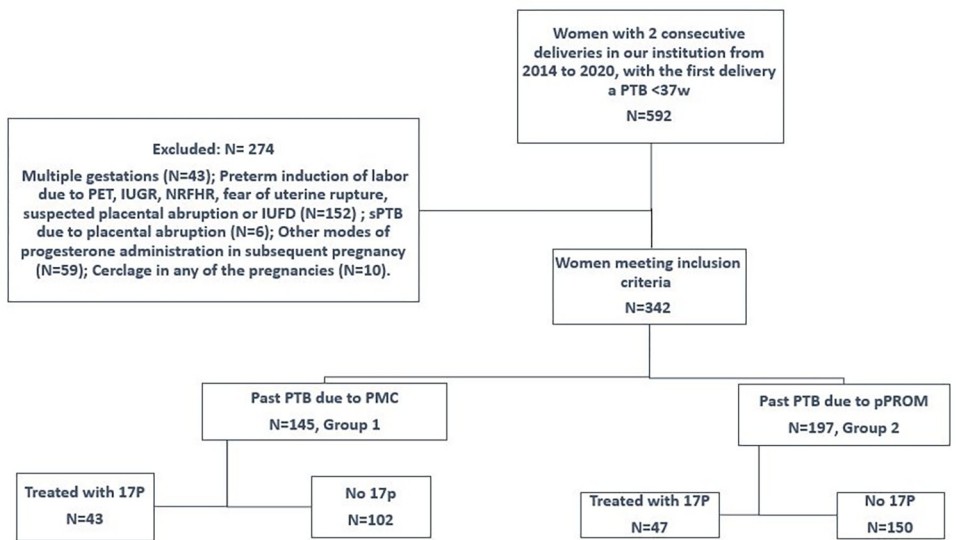

**Fig 1. Flowchart describing the study population.** Each group based on the presenting symptom–PMC or pPROM–was evaluated within itself comparing the treated vs. the untreated populations. Each woman was also analyzed individually according to the difference in GA between deliveries.

## Comparison between patients presenting with pPROM, treated or not treated with 17P during the subsequent pregnancy

There were no significant differences in basic characteristics between the treated and untreated populations (maternal age, ethnicity, background diseases, BMI, smoking, nulliparity, previous CD and previous cervical interventions) (Table 2).

## Individual benefit of 17P treatment

Given the above differences between treated and untreated populations in our cohort, we tried to overcome the selection bias of the treated population by evaluating the individual benefit of 17P treatment for each woman. Therefore, we calculated the delta between GA at the subsequent delivery and at the first preterm birth. The delta in each population indicated an extension of the subsequent pregnancy compared to the first preterm birth. These extensions were significantly greater in the treated population both for PMC (+4.3w vs. +2.6w in the treated vs. untreated group, respectively; p = 0.007) and for pPROM at the first sPTB (+3.7w vs. +2.7w in the treated vs. untreated groups, respectively; p = 0.018). This suggests a small individual benefit in 17P treatment, as gestation was longer in the subsequent delivery.

Logistic regression (Table 3) found two independent risk-factors for recurrent sPTB <35w: history of previous delivery < 35w (OR 3.25, CI 1.171–9.046) and admission during the current pregnancy due to concern regarding PTB (OR 4.325, CI 1.624–11.520). 17P treatment in the current pregnancy and progesterone treatment in the previous pregnancy were not found to be significant factors for recurrent PTB.

## Patients with a previous sPTB who presented with PMC compared to pPROM

Comparing the two presenting symptoms without regard to 17P treatment, PMC appeared to be the more pathological mechanism.

**Table 1. Previous PMC-PTB group comparing 17P treated population vs. untreated.**

| Variable* | Untreated (N = 102) | Treated (N = 43) | p-value |
|---|---|---|---|
| **Previous preterm delivery characteristics** | | | |
| Gestational diabetes | 7 (7.6%) | 3 (7%) | 0.896 |
| Intrauterine growth restriction | 7 (6.9%) | 2 (4.7%) | 0.614 |
| Hypertension | 4 (4%) | 0 (0%) | 0.186 |
| Progesterone (any kind) | 4 (3.9%) | 8 (19.5%) | 0.002 |
| Gestational age at delivery, weeks | 35.3 ± 2.6 | 33.4 ± 3.0 | <0.001 |
| Cesarean delivery | 16 (15.7%) | 7 (16.3%) | 0.929 |
| **Subsequent pregnancy and delivery characteristics** | | | |
| Interpregnancy interval, months | 18.4 ± 10.9 | 19.1 ± 10.0 | 0.720 |
| Gestational diabetes | 10 (11%) | 6 (14.3%) | 0.587 |
| Intrauterine growth restriction | 5 (4.9%) | 6 (14%) | 0.060 |
| Hypertension | 4 (4%) | 2 (4.7%) | 0.849 |
| Maternal-fetal specialist surveillance | 25 (24.5%) | 31 (72.1%) | <0.001 |
| Admission to hospital with PMC and suspected PTB | 11 (10.9%) | 13 (31.7%) | 0.003 |
| Cervical length at admission, mm | 24.5 ± 10.6 | 25.8 ± 9.4 | 0.770 |
| Cervical length at 14-16w, mm | 40.8 ± 5.9 | 35.4 ± 15.9 | 0.483 |
| Cervical length at 20-24w, mm | 39.2 ± 6.7 | 36.3 ± 6.3 | 0.145 |
| Gestational age at delivery, weeks | 37.9 ± 2.0 | 37.6 ± 2.1 | 0.460 |
| Gestational age delta from previous preterm delivery, w ± SD | +2.6 ± 3.3 | +4.3 ± 3.8 | 0.007 |
| PTB < 37w | 23 (22.5%) | 13 (30.2%) | 0.328 |
| PTB < 35w | 4 (3.9%) | 6 (14%) | 0.065 |
| PTB < 32w | 1 (1%) | 1 (2.3%) | 0.507 |
| Recurrent PMC causing PTB | 15 (14.7%) | 9 (20.9%) | 0.357 |
| pPROM in current pregnancy | 3 (2.9%) | 3 (7%) | 0.362 |
| Cesarean delivery | 13 (12.9%) | 11 (25.6%) | 0.061 |
| Birth weight, g | 2937.9 ± 471.2 | 2762.3 ± 534.8) | 0.051 |
| LBW | 12 (11.9%) | 13 (30.2%) | 0.008 |
| 5-min Apgar <7 | 2 (2%) | 0 (0%) | 1.000 |

*Values are presented as n (%) or mean ± standard deviation.

The treated group had an earlier previous sPTB (33.4w vs. 35.3w, p<0.001). During the subsequent pregnancy, more were considered to have a high-risk pregnancy and were treated by a MFM specialist (72.1% vs. 24.5%, p<0.001). They had a higher tendency to be admitted for observation with PMC and suspected sPTB (31.7% vs. 10.9%, p = 0.003). Finally, more patients in the treated group delivered <35 w (14% vs. 3.9%, p = 0.065).

A higher percentage of patients treated in the subsequent pregnancy had been already treated with progesterone in the prior pregnancy (19.5% vs. 3.9%, p = 0.002). Reasons for progesterone treatment in the prior pregnancy varied. In the treated group, 2 women (25%) received 17P due to a history of PTB, and 6 (75%) used vaginal progesterone due to short cervical length. In the untreated group, 2 women (50%) received 17P due to a history of PTB, and 2 (50%) used vaginal progesterone due to short cervical length.

No differences were found between the treated and untreated groups regarding rates of gestational diabetes (GDM), IUGR, HTN and CD in the first pregnancy.

Looking at the subsequent pregnancy, we found no differences in rates of sPTB < 32w, sPTB < 37w, mean GA at delivery, cervical lengths during pregnancy, GDM, HTN, interval between pregnancies and the presenting symptom of PTB, if it occurred.

A trend toward lower mean gestational weight was found in the treated group (2762g vs. 2937g, p = 0.051), with a higher rate of LBW < 2,500 g, (30.2% vs. 11.9%, p = 0.008). When correlating neonatal weight to GA, we found no differences in the rates of SGA (<10th percentile) or LGA (>90th percentile) between the groups. We found no differences between groups regarding Apgar scores and modes of delivery.

Basic maternal characteristics (age, ethnicity, background diseases, BMI, smoking, nulliparity, previous CD and previous cervical interventions) were similar between the PMC and pPROM groups (Table 4).

**Table 2. Previous pPROM-PTB group comparing 17P treated population vs. untreated.**

| Variable* | Untreated (N = 150) | Treated (N = 47) | p-value |
|---|---|---|---|
| **Previous preterm delivery characteristics** | | | |
| Gestational diabetes | 12 (8.6%) | 4 (8.7%) | 1.000 |
| Intrauterine growth restriction | 4 (2.6%) | 3 (6.4%) | 0.360 |
| Hypertension | 7 (4.7%) | 2 (4.3%) | 0.899 |
| Progesterone (any kind) | 1 (0.7%) | 4 (8.5%) | 0.013 |
| Gestational age at delivery, weeks | 35.7 ± 1.3 | 34.1 ± 2.5 | <0.001 |
| Cesarean delivery | 24 (15.9%) | 10 (21.3%) | 0.393 |
| **Subsequent pregnancy and delivery characteristics** | | | |
| Interpregnancy interval, months | 20.1 ± 13.5 | 20.3 ± 10.6 | 0.944 |
| Gestational diabetes | 12 (8.8%) | 4 (8.7%) | 1.000 |
| Intrauterine growth restriction | 8 (5.3%) | 0 (0%) | 0.202 |
| Hypertension | 3 (2%) | 2 (4.3%) | 0.337 |
| Maternal-fetal specialist surveillance | 19 (12.6%) | 33 (70.2%) | <0.001 |
| Admission to hospital with PMC and suspected PTB | 8 (5.4%) | 12 (26.1%) | <0.001 |
| Cervical length at admission, mm | 21.3 ± 18.0 | 25.2 ± 7.8 | 0.529 |
| Cervical length at 14-16w, mm | 41 ± 5.7 | 38.8 ± 5.5 | 0.458 |
| Cervical length at 20-24w, mm | 39.6 ± 6.6 | 38.3 ± 7.4 | 0.516 |
| Gestational age at delivery, w | 38.4 ± 1.5 | 37.7 ± 1.9 | 0.045 |
| Gestational age delta from preterm delivery, w ± SD | +2.7 ± 1.9 | +3.7 ± 2.6 | 0.018 |
| PTB< 37w | 22 (14.6%) | 12 (25.5%) | 0.082 |
| PTB< 35w | 2 (1.3%) | 5 (10.6%) | 0.009 |
| PTB< 32w | 0 (0%) | 0 (0%) | - |
| PMC in current pregnancy causing PTB | 10 (6.6%) | 3 (6.4%) | 1.000 |
| pPROM recured in current pregnancy causing PTB | 11 (7.3%) | 9 (19.1%) | 0.018 |
| Cesarean delivery | 24 (16.4%) | 6 (13%) | 0.580 |
| Birth weight, g | 3120.0 ± 528.0 | 3028.1 ± 574.1 | 0.309 |
| Apgar 5 min < 7 | 0 (0%) | 0 (0%) | - |

*Values are presented as n (%) or mean ± standard deviation.

The treated patients were more likely to have had a previous earlier delivery (34.1w vs. 35.7w, p<0.001) and to be treated by a MFM specialist during their subsequent pregnancy (70.2% vs. 12.6%, p<0.001). They also had more hospitalizations for PMC during the subsequent pregnancy (26.1% vs. 5.4%, p<0.001) and eventually a higher prevalence of sPTB < 35w (10.6% vs. 1.3%, p = 0.009) compared to the untreated group. Mean GA at delivery was lower in the treated group (37.7w vs. 38.4w, p = 0.045).

A higher percentage of patients treated in the subsequent pregnancy had been already treated with progesterone in the prior pregnancy (8.5% vs. 0.7%, p = 0.013). Reasons for progesterone treatment in the prior pregnancy varied. In the treated group, 3 women (75%) received 17P due to a history of PTB and 1 (25%) used vaginal progesterone due to short cervical length. In the untreated group, 1 woman (100%) had received 17P due to a history of PTB.

No differences were found between the treated and untreated groups in rates of GDM, IUGR or HTN in the first pregnancy.

Looking at the subsequent pregnancy, we found no differences in rates of sPTB < 32w, sPTB < 37w, cervical lengths during pregnancy, GDM, HTN, IUGR and interval between pregnancies comparing the groups. Recurrent pPROM in the subsequent pregnancy as the presenting symptom of another PTB was higher in the treated group (19.1 vs. 7.3% p = 0.018).

We found no differences between the groups regarding neonatal birth weights, rates of LBW, SGA, LGA, Apgar scores and modes of delivery.

# Discussion

## Principal findings

This study evaluated the individual benefit of 17P treatment in preventing recurrent sPTB, according to the presenting symptom of PMC or pPROM. We evaluated this benefit by calculating the difference in GA between the subsequent and previous deliveries, and found

**Table 3. Logistic regression: Risk factors for PTB <35w in subsequent delivery.**

| Variable | Odds Ratio | 95% CI | p-value |
|---|---|---|---|
| GA in previous delivery <35w | **3.26** | **1.17–9.05** | 0.024 |
| Admission in current pregnancy with PMC and suspected PTB | **4.33** | **1.62–11.52** | 0.003 |
| Progesterone treatment in previous pregnancy | 2.76 | 0.74–10.27 | 0.131 |
| 17P treatment in current pregnancy | 1.56 | 0.55–4.45 | 0.403 |

significantly greater difference in the treated population in both presenting symptoms, compared to the untreated population.

## Results

Previous studies have demonstrated that a history of sPTB is a risk-factor for recurrent sPTB in a subsequent pregnancy [3,4]. Our study found that the subsequent pregnancy of each woman treated with 17P was longer than her previous pregnancy. Although previous studies found that increasing parity itself might improve delivery outcomes since nulliparas are at higher risk for LBW and PTB [16,17], these studies did not evaluate the individual benefit of different treatments for each patient compared to herself, which is the novelty of the current study. In our cohort, as in previous reports, increasing parity itself prolonged the subsequence delivery, since a positive delta was also found in the untreated population in both PMC and pPROM groups. However, women treated with 17P had greater deltas, representing longer subsequent pregnancies compared to the untreated population. These findings might indicate that 17P extends the subsequent pregnancy beyond the expected extension contributed by parity itself and confers some benefit for all treated patients regardless of their initial presenting symptom.

The beneficial effect of 17P treatment in prolonging gestation might be explained by several mechanisms. Progesterone induces relaxation of the myometrial smooth muscle and blocks oxytocin action [18]. A local decrease in the progesterone level or the ratio of progesterone to estrogen is thought to have an important role in the initiation of spontaneous labor [7]. It would make sense that a higher level of progesterone with 17P treatment simply delays the natural course of preterm labor for each patient, resulting in an extension of her subsequent pregnancy.

In logistic regression, 17P treatment was not found significant in preventing recurrent sPTB <35w. This suggests there might be a small personal benefit in 17P treatment by prolonging the GA in the subsequent pregnancy, although this benefit probably did not decrease the rates of recurrent sPTB at thresholds of <32, <35 or <37w for the entire cohort. These results differ from those of Meis et al. [8] and agree with those of the PROLONG study [9]. This could be because our cohort was more similar to the population in the PROLONG study in terms of older mean GA at first PTB (33.4w ± 3.0 in the PMC group, 34.1w ± 2.5 in the pPROM group, 32.0w in the PROLONG study, and 30.6w ± 4.6 in Meis et al.); thus, demonstrating that 17P treatment was less effective for lower risk populations. With that said, we believe that in the era of personalized medicine, our approach of evaluating the individual benefit for each patient is preferred and should be investigated in other populations and in larger cohorts.

Comparing the two presenting symptoms–PMC and pPROM, without regard to 17P treatment, PMC appears to be the more pathological mechanism, with more women treated with progesterone and a lower mean GA in the preceding pregnancy. In the subsequent pregnancy, a higher percentage was treated by a MFM specialist, with a greater tendency to be admitted for observation for suspected sPTB.

**Table 4. Previous PTB PMC vs. previous pPROM.**

| Variable* | | PMC (N = 162) | pPROM (N = 216) | p-value |
|---|---|---|---|---|
| **Previous preterm delivery characteristics** | | | | |
| Gestational diabetes | | 11 (6.8%) | 19 (8.8%) | 0.303 |
| Intrauterine growth restriction | | 10 (6.2%) | 8 (3.7%) | 0.191 |
| Hypertension | | 4 (2.5%) | 9 (4.2%) | 0.274 |
| Progesterone treatment (any kind) | | 16 (9.8%) | 8 (3.7%) | 0.008 |
| Gestational age at delivery, weeks | | 34.6 ± 2.8 | 35.1 ± 1.9 | 0.024 |
| Cesarean delivery | | 29, (17.9%) | 37 (17.1%) | 0.475 |
| **Subsequent pregnancy and delivery characteristics:** | | | | |
| Interpregnancy interval, months | | 18.6 ± 10.1 | 20.2 ± 12.6 | 0.181 |
| Maternal-fetal specialist surveillance | | 63 (39%) | 60 (28%) | 0.015 |
| Gestational diabetes | | 18 (11.1%) | 18 (8.3%) | 0.231 |
| Intrauterine growth restriction | | 13 (8%) | 8 (3.8%) | 0.057 |
| Hypertension | | 7 (4.3%) | 6 (2.8%) | 0.296 |
| Admission to hospital with PMC and suspected PTB | | 32 (19.8%) | 23 (10.6%) | 0.010 |
| Cervical length at admission, mm | | 23.0 ± 10.0 | 22. 0 ± 13.0 | 0.750 |
| Cervical length at 14-16w, mm | | 38.0 ± 11.0 | 39.0 ± 7.0 | 0.683 |
| Cervical length at 20-24w, mm | | 37.0 ± 7.0 | 38.0 ± 8.0 | 0.417 |
| Gestational age at delivery, weeks | | 37.7 ± 2.1 | 38.1 ± 1.7 | 0.099 |
| PTB< 37w | | 42 (25.9%) | 43 (19.4%) | 0.085 |
| PTB< 35w | | 12 (7.4%) | 12 (5.6%) | 0.301 |
| PTB< 32w | | 3 (1.9%) | 1 (0.5%) | 0.212 |
| Recurrent PTB mechanism | PMC | 29 (69%) | 14 (33%) | <0.001 |
| | PPROM | 7 (19%) | 26 (60%) | |
| | Induction | 6 (14%) | 3 (7%) | |
| Cesarean delivery | | 27 (17.8%) | 33 (16.3%) | 0.407 |
| Gestational weight, g | | 2857.0 ± 499.0 | 3083.0 ± 547.0 | <0.001 |
| SGA | | 22 (13.6%) | 13 (6%) | 0.010 |
| LBW | | 31 (19.1%) | 27 (12.5%) | 0.052 |
| VLBW | | 4 (2.5%) | 2 (0.9%) | 0.219 |
| 5-min Apgar < 7 | | 7 (4.3%) | 4 (1.9%) | 0.135 |

*Values are presented as n (%) or mean ± standard deviation.

The PMC group had a relatively earlier previous sPTB but with minimal clinical significance (34.6w vs. 35.1w, p = 0.024). More women in the PMC group had been treated with progesterone in the prior pregnancy (9.8% vs. 3.7%, p = 0.008). During the subsequent pregnancy, more were considered to have a high-risk pregnancy and were treated by MFM specialist (39% vs. 28%, p = 0.015). They also had a greater tendency to be admitted for observation with PMC and suspected sPTB (19.8% vs. 10.6%, p = 0.010).

With that said, both groups eventually had similar rates of PTB <32, <35, <37w and mean GA at delivery. No differences were found regarding rates of GDM, HTN, cervical lengths during pregnancy, interpregnancy intervals and mode of delivery in the subsequent delivery.

Interestingly, although GA in the subsequent delivery was similar between groups, the PMC group had lower neonatal birth weights (2857g vs. 3083g, p<0.001) and higher rates of LBW (19.1% vs. 12.5%, p = 0.052 and SGA 13.6% vs. 6% SGA, p = 0.010) compared to the pPROM group.

The presenting symptom of sPTB in the first pregnancy tended to recur in subsequent pregnancies that ended with sPTB (69% in the PMC cohort, 60% in the pPROM cohort, p<0.001).

The higher rates of progesterone treatment in the preceding pregnancy in the PMC group are not surprising, since it is a well-known approach for cases of short cervix, which sometimes accompanies PMC. This presentation requires close surveillance and may lead to hospitalizations during pregnancy due to suspected PTB.

Previous studies have demonstrated relatively younger GA at delivery when PMC is the presenting symptom, compared to pPROM [13,14].

When comparing neonatal outcomes, although GA in subsequent deliveries were similar between the PMC and the pPROM groups, we found lower birth weights, and higher rates of LBW and SGA in the PMC group. These results suggest that the mechanism of PMC is related to smaller neonatal weights compared to that of pPROM. Previous studies described a relation between uterine contractions during pregnancy and higher fetal heart rates and heart preloads, suggesting uterine activity creates physiological challenges to the development and adaptation of the fetal cardiovascular system [19,20]. Other studies, evaluating pregnancies with PMC eventually delivering at term found lower neonatal birth weights compared to pregnancies without PMC [21,22].

Interestingly, whether PMC or pPROM, the presenting symptom tends to reoccur in cases of sPTB in the subsequent pregnancy (69% in the PMC cohort and 60% in the pPROM cohort, p<0.001). Recurrence of PTB mechanisms was also reported in previous studies [23,24].

We found that women who delivered very early during their first pregnancy tended to deliver early in their subsequent pregnancy as well. This has also been demonstrated in other studies showing that history of an earlier PTB is a risk-factor for recurrence of another early PTB [2,25,26]. As expected, and in agreement with previous data, we found two independent risk-factors for recurrent sPTB <35w: history of previous delivery <35w and admission in the current pregnancy for suspected sPTB. Logistic regression analysis results indicated that a previous delivery < 35w is an important risk-factor for recurrent PTB <35w.

In our cohort, only 39% of women in the PMC group and 28% in the pPROM group were treated by a MFM specialist in the following pregnancy and only 26.3% were treated prophylactically with 17P. These numbers are relatively low and attributed to low compliance with medical surveillance and lack of awareness of their high-risk status: Patients in the untreated population tended to have relatively long intervals between visits and were less likely to follow the ACOG guidelines for patients with a history of a sPTB and keep track of cervical length every 1–4 weeks starting at 16w GA [11]. Moreover, documentation of pregnancy visits lacked a discussion of the need for progesterone administration, implying that patients as well as doctors were unaware of patients' high-risk status.

Progesterone treatment in the previous pregnancy (representing chronic cervical shortening or a history of several PTBs) was not a significant risk-factor for recurrent sPTB <35w. Since a history of repeated PTB is a known risk-factor for PTB recurrence [27], we believe this risk-factor did not reach significance due to our relatively small cohort.

## Strengths and limitations

The strengths of the current study include its homogenous treatment, as data were collected from a single institution with the same strict management protocols. In addition, it included using only 17P as mode of treatment for prevention of recurrent PTB. Comparing the treated and untreated populations allowed to evaluate the overall effectiveness of 17P for each presenting symptom, which has not been considered previously. Women were compared to themselves to determine the individual effectiveness of 17P, a comparison that has not been considered previously.

This retrospective cohort study had some limitations. Due to its design, adherence with 17P could not be confirmed. Data regarding the latency between pPROM and onset of labor are also missing. Since we included only women who delivered twice in our institution, we could not investigate possible previous PTBs in other institutions. Thus, we could not reference a possible history of multiple PTBs. Excluding other presenting symptoms of PTB and other

modes of progesterone treatment created a relatively small cohort that might have been under-powered to demonstrate a decrease in the rates of recurrent sPTB at the different thresholds with 17P treatment. Our cohort had a selection bias, with a higher risk profile in the treated population, which affected the accuracy of the comparison between the two groups. However, calculating the delta GA between consecutive pregnancies for individual women and logistic regression analysis helped overcome this bias.

### Clinical implications

To the best of our knowledge, this is the first cohort to compare the efficacy of 17P for PTB to nontreatment due to presenting symptoms of PMC or pPROM, and to evaluate the individual benefit for each patient. We found an individual benefit from 17P for both conditions. We believe this information may assist physicians when debating whether to initiate 17P treatment for patients with a history of PTB. Information regarding the importance of previous GA as a risk factor for recurrent PTB <35w may also be helpful.

### Conclusions

According to our findings, the likelihood of a sPTB is increased in women with a previous sPTB <35w regardless of treatment with 17P during the subsequent pregnancy. The chances of having a repeat sPTB are higher for those with a previous sPTB that started with PMC. However, 17P treatment may prolong a pregnancy and consequently decrease potential adverse neonatal outcomes for all treated patients, regardless of their initial presenting symptom. Therefore, we cautiously propose that it might be justified to consider treatment with 17P for patients with a previous early sPTB, especially if it began with PMC.

### Supporting information

**S1 Table. Inclusion and exclusion criteria for the study.**
(DOCX)

**S1 Raw data. This is the raw data of our study cohort, with no identifiers.**
(XLSM)

### Acknowledgments

The authors thank Faye Schreiber, MS for editing the manuscript. She is an employee of Meir Medical Center.

### Author Contributions

**Conceptualization:** Gal Cohen, Tal Biron-Shental.

**Data curation:** Gal Cohen, Maya Shavit, Rimon Moran, Yael Yagur, Omer Weitzner, Michal Ovadia, Hanoch Schreiber.

**Formal analysis:** Netanella Miller.

**Investigation:** Maya Shavit, Rimon Moran, Yael Yagur, Omer Weitzner, Michal Ovadia, Hanoch Schreiber.

**Methodology:** Gal Cohen, Maya Shavit, Gil Shechter-Maor, Tal Biron-Shental.

**Project administration:** Gal Cohen.

**Writing – original draft:** Gal Cohen.

**Writing – review & editing:** Gil Shechter-Maor, Tal Biron-Shental.

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
