## [Decision Letter · Decision Letter 0]

8 Mar 2022

PONE-D-22-03037Can 17 hydroxyprogesterone caproate (17P) decrease preterm deliveries in specific patients?PLOS ONE

Dear Dr. Cohen,

Thank you for submitting your manuscript to PLOS ONE. After careful consideration, we feel that it has merit but does not fully meet PLOS ONE’s publication criteria as it currently stands. Therefore, we invite you to submit a revised version of the manuscript that addresses the points raised during the review process.

We look forward to receiving your revised manuscript.

Kind regards,

Frank T. Spradley

Academic Editor

PLOS ONE

Reviewers' comments:

Reviewer's Responses to Questions

**Comments to the Author**

1. Is the manuscript technically sound, and do the data support the conclusions?

Reviewer #1: Partly

Reviewer #2: Yes

2. Has the statistical analysis been performed appropriately and rigorously? 

Reviewer #1: Yes

Reviewer #2: Yes

3. Have the authors made all data underlying the findings in their manuscript fully available?

Reviewer #1: Yes

Reviewer #2: Yes

4. Is the manuscript presented in an intelligible fashion and written in standard English?

Reviewer #1: Yes

Reviewer #2: Yes

5. Review Comments to the Author

Reviewer #1: 

1. The title should have a clear and precise scientific meaning. “Specific patients” is not meaningful. A suggested (but no t required.

2. The authors should present to the audience all of the varied presenting symptoms of PTB and justify their selection of evaluating PMC and pPROM only in this study.

3. In Tables 1 and 2, what is mean by “gestational age delta from previous preterm delivery”? Does this refer to the change in when they delivered compared to the previous pregnancy? If so, there should be a + or – to indicate if the change is an extension or a reduction. This should also be included in the data presented in lines 228-230.

4. Lines275-282- How does this study distinguish the previous reported benefit of increasing parity to increase length of pregnancy, from their conclusion that 17P treatment increases length of pregnancy in these women?

5. Line 319- Please edit this sentence, it is unclear what the authors are trying to say.

6. Lines 328-329- How did the authors assess compliance with medical surveillance or determine if patients or doctors were aware of their high-risk status. These statements require justification and evidence to back them.

7. A discussion of the possible mechanisms that yield the individual benefit observed from 17P treatment would enhance the discussion.

Reviewer #2: Retrospective study to determine the benefit of 17P treatment in preventing reoccurrence of spontaneous preterm birth.

1) Abstract- Write the dose and via of administration of 17P

2) Introduction: Write the full names for ACOG and SMFM since that is the first time both names have been mentioned.

3) Provide information about samples size calculation. Did the authors calculate it?

4) A table with inclusion and exclusion criteria should be created.

5) Untreated groups were not receiving any medication or the untreated term was used just for those who did not received 17P?

6) Please add information about 17P and fetal outcomes from the previous studies mentioned (Meis and PROLONG study).

7) In 2012, ACOG recommended the prophylactic use of progesterone for prevention of preterm labor (PTL). Is there any recent ACOG update or change for the prophylactic use of progesterone for prevention of preterm labor after the PROLONG study? Please mention it if changes have been made.

8) Consider to add the reference below which compares relative effects of different types and routes of administration of progesterone preventing preterm birth. (https://doi.org/10.1111/1471-0528.15566).

6. PLOS authors have the option to publish the peer review history of their article (what does this mean?). If published, this will include your full peer review and any attached files.

Reviewer #1: No

Reviewer #2: No

---

## [Author Response · Author response to Decision Letter 0]

12 Apr 2022

Reviewer #1: 

1. A. The title should have a clear and precise scientific meaning. “Specific patients” is not meaningful. A suggested (but not required).

B. Thank you for your suggestion. We agree and have decided to change the title.

C. Page: 1, Lines: 1-3

D. Can 17 hydroxyprogesterone caproate (17P) decrease preterm deliveries in patients with a history of PMC or pPROM?

2. A The authors should present to the audience all of the varied presenting symptoms of PTB and justify their selection of evaluating PMC and pPROM only in this study.

B. Thank you for your review. We have chosen to evaluate PMC and pPROM since these are the two most common presenting symptoms of spontaneous PTB, and both have higher rates of recurrence. We have added a brief explanation of the different mechanisms of preterm labor and the reason for including only PMC and pPROM in our study cohort.

C: Pages: 5-7, Lines: 61-67, 75-81, 121-122 

D: "Preterm birth (PTB) accounts for 12.5% of births annually, and is a major cause of neonatal morbidity and mortality.1 Mechanisms leading to preterm birth include: Induction of labor due to maternal or fetal indications, accounting for 30-35% of PTB; premature contractions (PMC) with intact membranes, accounting for 40-45% of PTB; and preterm premature rupture of the membranes (pPROM) accounting for 20-25% of PTB. 2. Births that follow PMC and pPROM are also called spontaneous preterm births (sPTB)."

" Chorioamnionitis or placental abruption might also be involved in the mechanism of PTB. In these cases, maternal fever or vaginal bleeding, respectively, accompany the presenting symptom of PMC or pPROM. However – these ethnologies are relatively rare and are associated with lower rates of recurrence. 5,6 

Given the significant adverse outcomes associated with PTB, attempts to reduce the risk of recurrence have been made. Progesterone has been found to inhibit myometrial contraction,7 and so one strategy was the use of supplemental progestogens,"

" The current study evaluated the individual benefit of 17P treatment in preventing recurrent sPTB in the face of PMC and pPROM as the two main presenting symptoms. "

3. A In Tables 1 and 2, what is mean by “gestational age delta from previous preterm delivery”? Does this refer to the change in when they delivered compared to the previous pregnancy? If so, there should be a + or – to indicate if the change is an extension or a reduction. This should also be included in the data presented in lines 228-230.

B. Thank you for this enlightenment. The delta indeed refers to the change (weeks) in when they delivered compared to the previous pregnancy. All deltas calculated in our cohort indicated an extension of the current pregnancy, thus we added a + sign to all deltas, and an explanation in the results section.

C. Page:17, Lines: 253-258

D. "The delta in each population indicated an extension of the subsequent pregnancy compared to the first preterm birth. These extensions were significantly greater in the treated population both for PMC (+4.3w vs. +2.6w in the treated vs. untreated group, respectively; p=0.007) and for pPROM at the first sPTB (+3.7w vs. +2.7w in the treated vs. untreated groups, respectively; p= 0.018)."

4. Lines 275-282- How does this study distinguish the previous reported benefit of increasing parity to increase length of pregnancy, from their conclusion that 17P treatment increases length of pregnancy in these women?

B: Thank you for your review. The fact that a greater delta was found in the treated population compared to the untreated, implies that 17p extended the subsequent pregnancy beyond the expected extension contributed by parity itself. We added this explanation to the discussion section. 

C: Page: 21-22 Lines: 306-313

D: "In our cohort, as in previous reports, increasing parity itself prolonged the subsequence delivery, since a positive delta was also found in the untreated population in both PMC and pPROM groups. However, women treated with 17P had greater deltas, representing longer subsequent pregnancies compared to the untreated population. These findings might indicate that 17P extends the subsequent pregnancy beyond the expected extension contributed by parity itself and confers some benefit for all treated patients regardless of their initial presenting symptom. "

5. A. Line 319- Please edit this sentence, it is unclear what the authors are trying to say.

B. Thank you for this comment, we apologize for this omission and have made the corrections needed to clarify this sentence. 

C. Page: 24. Lines: 357-358

D: " We found that women who delivered very early during their first pregnancy tended to deliver early in their subsequent pregnancy as well. " 

6. Lines 328-329- How did the authors assess compliance with medical surveillance or determine if patients or doctors were aware of their high-risk status. These statements require justification and evidence to back them.

B. Thank you for your comment. Since all patients in the cohort delivered in our institution, we were able to collect data of their pregnancy follow up.

Patients in the untreated population tended to have relatively long intervals between visits and were less likely to follow the ACOG guidelines for patients with a history of a sPTB and keep track of cervical length every 1-4 weeks starting at 16w GA (ACOG practice bulletin 234, PMID: 34293771). The fact that pregnancy follow-up documentation lacked a discussion of the need for progesterone administration implies that patients as well as the doctors were unaware of patients' high-risk status. We added this justification to our manuscript in the discussion. 

C. Page: 24 Lines: 370-375

D. " Patients in the untreated population tended to have relatively long intervals between visits and were less likely to follow the ACOG guidelines for patients with a history of a sPTB and keep track of cervical length every 1-4 weeks starting at 16w GA.8 Moreover, documentation of pregnancy visits lacked a discussion of the need for progesterone administration, implying that patients as well as the doctors were unaware of patients' high-risk status."

7. A discussion of the possible mechanisms that yield the individual benefit observed from 17P treatment would enhance the discussion.

B. We appreciate your suggestion, and we have added such a discussion under our discussion section.

C. Page: 22. Lines: 314-320

D: "The beneficial effect of 17P treatment in prolonging gestation might be explained by several mechanisms. Progesterone induces relaxation of the myometrial smooth muscle and blocks oxytocin action. A local decrease in the progesterone level or the ratio of progesterone to estrogen is thought to have an important role in the initiation of spontaneous labor. It would make sense that a higher level of progesterone with 17P treatment simply delays the natural course of preterm labor for each patient, resulting in an extension of her subsequent pregnancy."

Reviewer #2: Retrospective study to determine the benefit of 17P treatment in preventing reoccurrence of spontaneous preterm birth.

1) A. Abstract- Write the dose and via of administration of 17P

B. Thank you for this comment, we added the dose and mode of administration

C. Page: 3 Lines: 34-35

D. "During the subsequent pregnancy, 90 (26.3%) patients received 250 mg 17P IM"

2) A. Introduction: Write the full names for ACOG and SMFM since that is the first time both names have been mentioned.

B. We thank you for this enlightenment, we have added the full names.

C. Page: 6, Lines: 88-92

D. " This made 17P the preventive treatment of choice by the American College of Obstetricians and Gynecologists (ACOG) and the Society for Maternal-Fetal Medicine (SMFM), until the recent PROLONG study was published, showing 17P had no benefit in decreasing rates of recurrent PTB or neonatal morbidity.9 

3) A. Provide information about samples size calculation. Did the authors calculate it?

B. Thank you for your comment. Power analysis was made under the assumption that the incidence for recurrent PTB will be 54.9% for the untreated population and 36.3% with 17P treatment (Meis. Et al PMID: 12802023, DOI:10.1056/NEJMoa035140). The sample size calculation indicated at least 111 participants were needed for each arm of the study, with an alpha of 0.05 and 80% power." Eventually, due to patient's low compliance with medical surveillance and lack of awareness of their high-risk status, only 39% of women in the PMC group and 28% in the pPROM group were treated by a MFM specialist in the following pregnancy and only 26.3% were treated prophylactically with 17P. That created a smaller cohort than expected, and must have been the reason why 17P's beneficial effect was only demonstrated by prolongation of the GA in the subsequent pregnancy (greater deltas for the treated population) and not by a decrease in the rates of recurrent sPTB at different thresholds <32, <35 or <37w. Our study focused on the personal benefit of 17P represented by the prolongation of the GA in each woman compared to herself (represented by the delta), and that outcome reached the level of significance. 

We have added this limitation to the discussion section

C: Page: 25 Lines: 393-396

D: " Excluding other presenting symptoms of PTB and other modes of progesterone treatment created a relatively small cohort that might have been underpowered to demonstrate a decrease in the rates of recurrent sPTB at the different thresholds with 17P treatment."

4) A. A table with inclusion and exclusion criteria should be created.

B. Thank you for this review, we have added this table to our manuscript (Supplementary table 1)

C. Page: 10 Lines 187-188

D. " Supplementary table 1 shows the inclusion and exclusion criteria for the study"

Inclusion criteria Exclusion criteria

Singleton pregnancies Multiple gestations

Patients with 2 consecutive deliveries in our institution with the first delivery at < 37w GA Preterm induction of labor due to PET, IUGR, NRFHR, fear of uterine rupture, suspected placental abruption or IUFD

 Spontaneous preterm birth due to placental abruption

 Patients administered with other modes of progesterone in subsequent pregnancy (Vaginal, PO) 

 Cerclage in any of the pregnancies

 Pessary in any of the pregnancies

5) Untreated groups were not receiving any medication or the untreated term was used just for those who did not receive 17P?

B. Thank you for this clarification. Untreated groups were not receiving any kind of progesterone (IM, PO, vaginal) nor any other medication to prevent preterm birth. We have added this explanation to the methods section.

C. Page: 8 Lines: 129-134

D: " Each group was further divided into 2 subgroups: treated with 17P in the subsequent pregnancy – including all women treated with weekly 250 mg 17P IM starting at 16 weeks of gestation, and untreated - those not treated with any kind of progesterone (vaginal, PO, IM) or any other medication to prevent sPTB in the subsequent pregnancy."

6) A. Please add information about 17P and fetal outcomes from the previous studies mentioned (Meis and PROLONG study).

B. Thank you for your review. Meis et. Al showed that Infants of women treated with 17P had significantly lower rates of necrotizing enterocolitis, intraventricular hemorrhage, and need for supplemental oxygen. The PROLONG study found no difference in neonatal morbidity index between the treated and untreated population. We have added this information to the introduction section.

C: Page: 6, Lines: 83-92 

D: " This treatment was evaluated in two large randomized-controlled-trials. Meis et al.5 demonstrated a substantial decrease in the rate of recurrent PTB with 17P treatment compared to placebo, among high-risk populations, as well as a decrease in neonatal adverse outcomes such as necrotizing enterocolitis, intraventricular hemorrhage, and need for supplemental oxygen. This made 17P the preventive treatment of choice by the American College of Obstetricians and Gynecologists (ACOG) and the Society for Maternal-Fetal Medicine (SMFM), until the recent PROLONG study was published, showing no17P had no benefit in decreasing rates of recurrent PTB or neonatal morbidity.6"

7) In 2012, ACOG recommended the prophylactic use of progesterone for prevention of preterm labor (PTL). Is there any recent ACOG update or change for the prophylactic use of progesterone for prevention of preterm labor after the PROLONG study? Please mention it if changes have been made.

B: Thank you very much, we appreciate your comment. The ACOG referred to this issue in their last practice bulletin on August 2021, recommending progesterone supplementation (either vaginal or IM) for patients with a history of sPTB. We have added their statement to our manuscript. 

C: Page: 6, Lines: 102-195

D. " On August 2021, the ACOG stated that patients with a singleton pregnancy and a prior sPTB should be offered progesterone supplementation (either vaginal or IM) in the context of a shared decision-making process incorporating the available evidence and the patient's preferences.7"

8) A. Consider to add the reference below which compares relative effects of different types and routes of administration of progesterone preventing preterm birth. (https://doi.org/10.1111/1471-0528.15566).

B. Thank you for the suggestion, we have added this reference to our manuscript in the introduction section.

C. Page: 6, Lines: 98-101

D. " Vaginal progesterone also has been studied for the indication of prior PTB. A systematic review and meta-analysis7 comparing the relative effects of different kinds of progesterone in preventing PTB, concluded that both vaginal progesterone and 17P were effective at reducing the risk for recurrent PTB."

---

## [Decision Letter · Decision Letter 1]

29 Apr 2022

Can 17 hydroxyprogesterone caproate (17P) decrease preterm deliveries in patients with a history of PMC or pPROM?

PONE-D-22-03037R1

Dear Dr. Cohen,

We’re pleased to inform you that your manuscript has been judged scientifically suitable for publication and will be formally accepted for publication once it meets all outstanding technical requirements.

Kind regards,

Frank T. Spradley

Academic Editor

PLOS ONE

Reviewers' comments:

Reviewer's Responses to Questions

**Comments to the Author**

1. If the authors have adequately addressed your comments raised in a previous round of review and you feel that this manuscript is now acceptable for publication, you may indicate that here to bypass the “Comments to the Author” section, enter your conflict of interest statement in the “Confidential to Editor” section, and submit your "Accept" recommendation.

Reviewer #2: All comments have been addressed

2. Is the manuscript technically sound, and do the data support the conclusions?

Reviewer #2: Yes

3. Has the statistical analysis been performed appropriately and rigorously? 

Reviewer #2: Yes

4. Have the authors made all data underlying the findings in their manuscript fully available?

Reviewer #2: Yes

5. Is the manuscript presented in an intelligible fashion and written in standard English?

Reviewer #2: Yes

6. Review Comments to the Author

Reviewer #2: (No Response)

7. PLOS authors have the option to publish the peer review history of their article (what does this mean?). If published, this will include your full peer review and any attached files.

Reviewer #2: No

---

## [Editor Report · Acceptance letter]

4 May 2022

PONE-D-22-03037R1 

Can 17 hydroxyprogesterone caproate (17P) decrease preterm deliveries in patients with a history of PMC or pPROM? 

Dear Dr. Cohen:

I'm pleased to inform you that your manuscript has been deemed suitable for publication in PLOS ONE. Congratulations! Your manuscript is now with our production department. 

Kind regards, 

on behalf of

Dr. Frank T. Spradley 

Academic Editor

PLOS ONE